# FUSE-ing Language Models: Zero-Shot Adapter Discovery for Prompt Optimization Across Tokenizers

**Joshua Nathaniel Williams**
Department of Computer Science
Carnegie Mellon University
Pittsburgh, PA 15213, USA
jnwillia@cs.cmu.edu

**J. Zico Kolter**
Department of Machine Learning
Carnegie Mellon University
Pittsburgh, PA 15213, USA
zkolter@cs.cmu.edu

## Abstract

The widespread use of large language models has resulted in a multitude of tokenizers and embedding spaces, making knowledge transfer in prompt discovery tasks difficult. In this work, we propose FUSE (Flexible Unification of Semantic Embeddings)[1], an inexpensive approach to approximating an adapter layer that maps from one model's textual embedding space to another, even across different tokenizers. We introduce a third-order tensor-based representation of a model's embedding space that aligns semantic embeddings that have been split apart by different tokenizers, and use this representation to derive an approximation of the gradient of one model's outputs with respect to another model's embedding space. We show the efficacy of our approach via multi-objective optimization over vision-language and causal language models for image captioning and sentiment-based image captioning.

## 1 Introduction

The current popularity of large language models (LLMs) has led to many individuals and organizations training and fine-tuning models for their own needs, resulting in a myriad of models with unique ways of processing, tokenizing, and embedding text. This diversity creates a challenge for knowledge transfer and interoperability across models, effectively siloing the insights and capabilities of any single model. One popular way of enabling interoperability is through *prompting* strategies. These approaches leverage the ability for text to be passed across models, by converting tasks into formats that LLMs can solve. However, the uniqueness of different models' token and embedding spaces creates difficulties in automated methods for prompt discovery.

While prompting strategies have found success across a variety of tasks including adversarial text generation (Zou et al., 2023), text summarization (Zhang et al., 2022), and prompt discovery for generative models (Wen et al., 2024), the non-differentiable nature of text remains a limitation. One way of addressing this challenge is by encouraging a standardized tokenization and embedding strategy, where every new model or architecture uses the same tokenizer and embedding space. Despite the potential for fostering cooperation across models, it is unlikely that model developers will converge on a single tokenization. Yet, such a standardized representation may not be necessary if we can freely compute forward and backward passes across models, regardless of their tokenization.

In our work, we propose one such method of computing gradients across different models' discrete embedding spaces, even if these spaces are defined in terms of different tokenizers. Our approach, which we call *FUSE* (Flexible Unification of Semantic Embeddings) inserts a simple module that approximates the functionality of an adapter layer that maps between the embeddings of multiple models without finetuning. We find that rather than focusing on individual tokens, if we instead focus on groups of tokens separated by whitespace, then

---

[1] https://github.com/jnwilliams/FUSE_prompt_inversion.git

we can track how a full word is represented in the embeddings space and create a necessary equivalence among tokenizers that can be leveraged to map from one model to another. We then derive a strategy to compute one such differentiable map, and find that we can approximate the gradient of a language model's output with respect to another model's embedding space solely in terms of the first model's embedding and a precomputed tensor.

We show the effectiveness of our approach through a zero-shot captioning task and zero-shot captioning with sentiment, where each task is solved via a multi-objective optimization over the sum of the models' losses. Our contributions are as follows: 1) We introduce a new framing for optimizing problems across embedding spaces that focus on groups of whitespace-separated tokens, rather than individual tokens. 2) We show how to compute an approximate gradient through the input embedding spaces of different models, even in the case where the tokenizer and vocabulary for each model varies significantly. 3) We show how we can enable zero-shot tasks by composing multiple specialized models with no additional finetuning, through image captioning tasks.

## 2 Background and Related Work

### 2.1 Prompt Engineering and Discovery

Prompting has become a very useful tool for unlocking the knowledge of large, pretrained language models. By carefully crafting prompts to LLMs, we can find simple strategies that can be used to guide the model toward a specific task. For example, Radford et al. (2019) have framed the task of text summarization as an LLM task by appending "TLDR:" at the end of an article and then having the model generate the text that best follows.

Prompt engineering and discovery (Chen et al., 2023; Gu et al., 2023) focus on finding effective prompts for a variety of tasks. Early approaches, such as AutoPrompt (Shin et al., 2020) used a gradient-based search strategy to discover appendable suffixes for a prompt that guide the model to act as a sentiment classifier on its original input. FluentPrompt (Shi et al., 2022) improved upon this by applying a language prior on the suffix, better aligning prompt discovery with more human-like prompts. This approach has also been successful as an adversarial attack on aligned models. Zou et al. (2023) have introduced Greedy Coordinate Gradients (GCG), to discovers suffixes that can be appended to a harmful prompt in order to circumvent safety measures in the LLM and generate harmful responses. Additional work (Zhu et al., 2023; Chao et al., 2023) further built on these attacks in order to efficiently find adversarial prompts to elicit harmful behaviors in the models.

These strategies extend to generative image models. Wen et al. (2024) leverage CLIP (Radford et al., 2021), to find prompts that align well with an image, enabling them to "invert" the image generation process. In contrast to other search methods (Zou et al., 2023; Shin et al., 2020), the authors introduce an approach that finds prompts via a form of projected gradient descent. Mahajan et al. (2023) use similar projected gradient descent methods to optimize prompts directly through the diffusion process of a diffusion model. Their approach has found prompts more closely tailored to a specific generative process. ClipCAP (Mokady et al., 2021) and ZeroCAP (Tewel et al., 2022), have found additional success in image captioning by finetuning and optimizing aspects of pretrained language models in order to better align their output with the CLIP similarity of the prompt with the image.

### 2.2 Prompt Discovery with Knowledge Transfer

While automated prompt discovery for a single model is powerful, we can both broaden the number of applicable tasks and improve upon existing methods by allowing multiple systems to exchange information (Geraci, 1991; Nilsson, 2019; Hu et al., 2023). For example, Chao et al. (2023) have found that using a discriminator to determine the degree of success for an attack in tandem with other prompt discovery approaches can find successful natural language prompts that elicit harmful behavior faster than alternative apporaches.

We focus our work on knowledge transfer from one model to another by centering our attention on the text embedding layers of language models. In contrast to the above methods,

our work aligns with prior work on adapter models, in which a new layer is inserted between layers of pretrained models (He et al., 2021; Houlsby et al., 2019; Bapna et al., 2019). These layers are then finetuned, adapting the model to new tasks, including knowledge transfer between multiple models (Wang et al., 2020). By inserting a new layer just before the embedding layer of a model that approximates the behavior of an adapter from one model's embedding space to another, we can make use of a weighted combination models to solve tasks without requiring a specific architecture nor requiring additional fine-tuning. By allowing the gradients to flow freely across models with different tokenizers and embedding spaces, we can optimize prompts for zero-shot tasks leveraging the knowledge within multiple models with relatively little overhead.

## 3 Methodology

Before going into detail on our approach, we first introduce key concepts and notation that will be necessary for understanding our method. Throughout this work, scalars, vectors, matrices, and tensors are denoted as lowercase, $a$, bolded lowercase $\mathbf{a}$, uppercase $A$, and as uppercase with a tilde $\tilde{A}$, respectively.

### 3.1 Language Model Embeddings

Given a string, a tokenizer maps it to a set of tokens, $\mathbf{t} \in \{0, ..., |V|\}^s$, where $s$ is the length of the tokenized string and $|V|$ is the number of unique tokens in the tokenizer. The model then applies a mapping $\mathcal{E} : \{0, ..., |V|\}^s \rightarrow \mathbb{R}^{s \times d}$ which indexes these tokens across a discrete set, mapping each to a unique $d$-dimensional embedding, $E \in \mathbb{R}^{s \times d}$.

Alternatively, we can represent the embedding function ($\mathcal{E}$) itself as a matrix, $V \in \mathbb{R}^{|V| \times d}$, where each row corresponds to the embedding vector for a specific token in the vocabulary. By representing the tokens as one-hot encodings over the vocabulary, $X \in \{0, 1\}^{s \times |V|}$, we can express the embedding vectors with a lookup operation $E = XV$. In this framing, $V$ is both a matrix and denotes the set of discrete embedding vectors for a model.

With this set of preliminary information in hand, we proceed to outline our approach, starting from the simple case in which models share a tokenizer, but have different embeddings (i.e., strings will always be tokenized to the same $\mathbf{t}$, but the embedding mapping, $\mathcal{E}(\mathbf{t})$ differs between models. We then build on this case and extend it to the case in which models tokenize words differently and have different embedding mappings (i.e., words may be separated arbitrarily, the model vocabularies have different lengths, and each embedding may have a different dimensionality across models).

For the latter case, understanding how to multiply tensors is crucial for our approach. When working with tensors of order greater than 2, their multiplication has been well-defined in terms of the t-product operator, $*$ (Kilmer & Martin, 2011). The t-product defines an associative and left/right distributive multiplication operation of $\tilde{A} \in \mathbb{R}^{m \times k \times p_1 \times \cdots \times p_n}$ and $\tilde{B} \in \mathbb{R}^{k \times n \times p_1 \times \cdots \times p_n}$, where $\tilde{A} * \tilde{B} \in \mathbb{R}^{m \times n \times p_1 \times \cdots \times p_n}$. We also make use of the folding and unfolding operation introduced alongside the t-product that reshapes an $\mathbb{R}^{d_1 \times d_2 \times \cdots \times d_n}$ tensor into a partitioned tensor in $\mathbb{R}^{d_1 d_n \times \cdots \times d_{n-1}}$ tensor and back,

$$\text{unfold}(\tilde{X}) = \begin{bmatrix} \tilde{X}_1 & \tilde{X}_2 & \cdots & \tilde{X}_n \end{bmatrix}^T \qquad \text{fold}(\text{unfold}(\tilde{X})) = \tilde{X}.$$

Note that Kilmer & Martin (2011) require, $\tilde{A}$ and $\tilde{B}$ to have their first two dimensions of the appropriate shape for matrix multiplication and each of the remaining dimensions must be the same size, however this product can also be generalized to arbitrary tensor sizes as long as the first two dimensions are appropriate sizes for matrix multiplication. See Appendix A for a further primer on the t-product and this generalization.

The key idea in our work is that while current tokenizers may split the same word arbitrarily, they always respect white-space separation. We can build shared representations across embedding spaces by focusing on groups of white-space separated tokens and their

embeddings, represented as third order tensors, rather than individual tokens and embeddings represented by matrices. In doing so, we find that we can approximate the gradient of a language model's output with respect to another model's embedding space solely in terms of the first model's embedding of a string and a precomputed tensor.

## 3.2 Shared Tokenizers

Recall that the embedding of a set of tokens for model $i$, can be represented as, $E_i = XV_i$, where $X$ is a one-hot encoding across the vocabulary, $V_i$ [2]. Our goal is to solve a multi-objective optimization over $K$ models, in which each model is solving a different task whose loss is computed with a differentiable $\mathcal{L}_i(E_i)$.

$$\arg\min_X \sum_{i=1}^K \mathcal{L}_i(XV_i). \tag{1}$$

As each model uses the same tokenizer, $X$ is shared for each model. This problem can clearly be solved via any off-the-shelf optimizer. However, consider the pedagogical case in which we want to directly optimize the embedding vectors, $E_i$, instead of the one-hot encodings. Solving equation (1) becomes less clear. One approach is to choose one model to be the *primary model*, and use its embeddings as input to all other models by introducing an adapter $\mathcal{T}_{i:j} : V_i \to V_j$ that maps from model $i$'s vocabulary to model $j$'s vocabulary. With the introduction of $\mathcal{T}_{i:j}$, we only optimize in the primary model's embedding space and our objective becomes,

$$\arg\min_{E_i} \mathcal{L}_i(E_i) + \sum_{j\neq i} \mathcal{L}_j(\mathcal{T}_{i:j}(E_i)). \tag{2}$$

With a differentiable representation of $T_{i:j}$, then this equation can be solved via gradient-based optimization. However, as the vocabulary matrices are not square, they are not invertible; we cannot directly map from the embedding space to back to token space. We instead approximate a linear map for $\mathcal{T}_{i:j}$ using the Moore-Penrose inverse (pseudoinverse) of the model's vocabulary, $V_i^+ = V_i^T(V_iV_i^T)^{-1}$. By using the pseudoinverse, $E_iV_i^+ \approx X$, we can substitute $E_iV_i^+$ for every instance of $X$ in Equation (1) and set $\mathcal{T}_{i:j}(E_i) = E_j \approx E_iV_i^+V_j$. The gradient of Equation (2), is then a simple application of the chain rule,

$$\nabla_{E_i}\mathcal{L}_j(\mathcal{T}_{i:j}(E_i)) \approx \left(\nabla_{E_j}\mathcal{L}_j(E_j)\right)V_i^+V_j. \tag{3}$$

Pay particular attention to the fact that the approximate gradient is no longer dependent on the embedding of the model that we want to map *from*, only on the embedding that we want to map *to*. We can thus map $E_i$ to $E_j$ in a non-differentiable way (e.g., convert back to text and retokenize), compute the gradient of the loss for model $j$, with respect to its own embeddings, and then multiply this gradient by $V_i^+V_j$ to approximate the gradient of the loss of *any* secondary model with respect to the embeddings of the primary model. This enables us to freely have access to noisy descent methods across a variety of models and zero-shot tasks, while only keeping track of a single $d_i \times d_j$ matrix per additional model.

## 3.3 Different Tokenizers

While the previous section enables gradient-based methods directly on the embedding space, it relies on models tokenizing words in the same way. For example, if we tokenize the word "Happy", equation (3) assumes that the $k$-th token in every model's vocabulary is the embedding for "Happy". But when using different tokenizers, this is no longer true. If one model tokenizes the word "Happy" as {'Ha','ppy'} and another as a single token, {'Happy'}, equation (3) gives incompatibly sized gradients in $\mathbb{R}^{2\times d}$ and $\mathbb{R}^{1\times d}$. The primary question becomes: "How do we reconcile these incompatible gradients?"

---

[2]Note that $V_i$ uses the subscript $i$ to denote the vocabulary of a particular model, not the token index within a model's vocabulary.

$$
\tilde{V}_i = \begin{bmatrix} [\ \mathbf{the}\ ] & , & [\ \varnothing\ ] \\ [\ \mathbf{qui}\ ] & , & [\ \mathbf{ck}\ ] \\ [\ \mathbf{br}\ ] & , & [\ \mathbf{own}\ ] \\ [\ \mathbf{fox}\ ] & , & [\ \varnothing\ ] \\ [\ \mathbf{j}\ ] & , & [\ \mathbf{umps}\ ] \\ [\ \mathbf{over}\ ] & , & [\ \varnothing\ ] \\ [\ \mathbf{the}\ ] & , & [\ \varnothing\ ] \\ [\ \mathbf{l}\ ] & , & [\ \mathbf{azy}\ ] \\ [\ \mathbf{dog}\ ] & , & [\ \varnothing\ ] \end{bmatrix} \qquad \tilde{V}_j = \begin{bmatrix} [\ \mathbf{the}\ ] & , & [\ \varnothing\ ] \\ [\ \mathbf{q}\ ] & , & [\ \mathbf{uick}\ ] \\ [\ \mathbf{brown}\ ] & , & [\ \varnothing\ ] \\ [\ \mathbf{fox}\ ] & , & [\ \varnothing\ ] \\ [\ \mathbf{jump}\ ] & , & [\ \mathbf{s}\ ] \\ [\ \mathbf{ov}\ ] & , & [\ \mathbf{er}\ ] \\ [\ \mathbf{the}\ ] & , & [\ \varnothing\ ] \\ [\ \mathbf{lazy}\ ] & , & [\ \varnothing\ ] \\ [\ \mathbf{d}\ ] & , & [\ \mathbf{og}\ ] \end{bmatrix}
$$

Figure 1: An $\mathbb{R}^{9 \times d \times 2}$ tensor vocabulary over words: "the quick brown fox jumps over the lazy dog". Each plain-text word represents its corresponding $\mathbb{R}^d$ embedding, and each $\varnothing$ is a 0 vector. We approximate the gradient for a mapping from model $\mathcal{M}_i$'s embeddings to $\mathcal{M}_j$'s embeddings by computing the t-product $\tilde{V}_i^+ * \tilde{V}_j$, where $\tilde{V}_i^+$.

Consider a case in which we split a string into a batch of its component white-space separated words and then compute the gradient of some function over each word in the batch. Even if words are tokenized differently, the total derivative with respect to a word's multi-token representation still provides information on a loss-minimizing direction.

We therefore propose an embedding representation that focuses on batches of words, rather than individual tokens, by introducing split and merge [3] operations analogous to the fold and unfold operations used by Kilmer & Martin (2011) when defining the t-product.

$$\text{split}(E) = \begin{pmatrix} \tilde{E}_1 & \tilde{E}_2 & \cdots & \tilde{E}_k \end{pmatrix} \qquad \text{merge}(\text{split}(E)) = E,$$

where $\tilde{E}_i \in \mathbb{R}^{1 \times d \times l_i}$ is the third-order tensor representation of the a set of tokens in $E$, and $l_i$ are the number of tokens that make up the word represented by $\tilde{E}_i$. The split operation does not return a tensor (denoted by the change from brackets to parenthesis) but a list of tensors where each element is a whitespace-separated set of tokens in the original string that can have variable length, $l_i$ [4]. The merge operation stacks these tensors back into their original shape. Using the limited vocabulary in Figure 1 (and denoting each embedding vector in $\mathbb{R}^d$ as the plain-text token that it represents), calling 'split' on an embedding, $\epsilon \in \mathbb{R}^{6 \times d}$ that represents the phrase: "the quick brown fox", gives

$$\text{split}(\epsilon) = \left( [[\mathbf{the}]] \quad \begin{bmatrix} [\mathbf{qui}] \\ [\mathbf{ck}] \end{bmatrix} \quad \begin{bmatrix} [\mathbf{br}] \\ [\mathbf{own}] \end{bmatrix} \quad [[\mathbf{fox}]] \right).$$

Using this lens, we extend the second-order vocabulary tensor to a third-order tensor, $\tilde{V} \in \mathbb{R}^{w \times l \times d}$, where $w$ are the number of words that that can be represented by the original vocabulary $V$ using at most $l$ tokens. Any set of tokens that requires fewer than $l$ tokens to represent is assumed zero-padded. See Figure 1 for an example of $\tilde{V}$ across two models.

Importantly, Jin et al. (2017) have shown the Moore-Penrose inverse still exists for arbitrary tensors under the t-product. We can therefore reuse the ideas in section 3.2, however, rather than matrix multiplication, we instead use the tensor t-product. If the embedding for a word is represented as

$$\tilde{E} = \tilde{X} * \tilde{V} \qquad \tilde{X} = \text{fold}(\begin{bmatrix} X & 0 & \cdots & 0 \end{bmatrix}),$$

where $\tilde{E} \in \mathbb{R}^{s \times d \times l}$, $\tilde{X} \in \{0,1\}^{s \times |V| \times l}$ is the one-hot tensor encoding for the t-product and $X \in \{0,1\}^{s \times |V|}$ is the matrix one-hot encoding. We can construct $\tilde{E}_i$ and $\tilde{E}_j$ with a system of equations and follow the same process from Section 3.2 to compute a differentiable

---

[3]For clarity, we simplify the split and merge operations throughout this section. Each split and merge are specific to a model and both have access to the original string that the embeddings represent. A more formal notation may be, $\text{split}_S^i(E)$, however this may introduce unnecessary confusion for the reader. Throughout 3.3, assume that split and merge have all necessary information to shape tensors into their appropriate shapes for each operation.

[4]For convenience, we also define the split operation to be distributive for any arbitrary function, except for the merge function that acts as an inverse. $f(\text{split}(E)) = \begin{pmatrix} f(\tilde{E}_1) & f(\tilde{E}_2) & \cdots & f(\tilde{E}_k) \end{pmatrix}$.

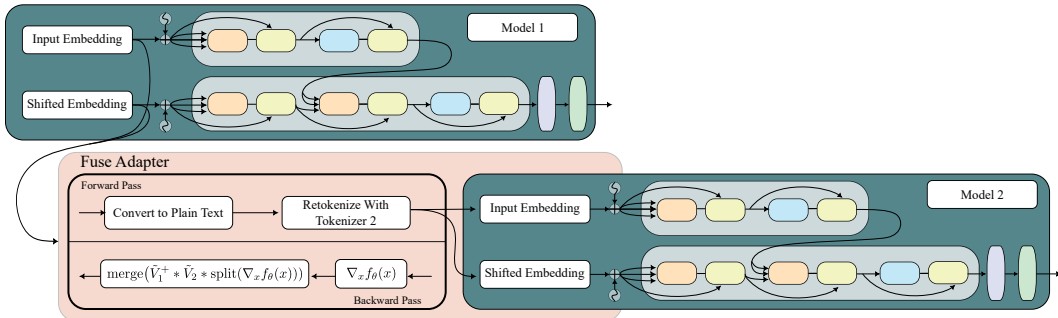

Figure 2: The FUSE adapter connecting two transformer models for parallel inference. Inputs from Model 1 flow through the adapter by converting to text, retokenizing with Model 2's tokenizer, and embedding into Model 2's input space. The backward pass receives the gradient from Model 2, and multiplies it by the precomputed $\tilde{V}_1^+ * \tilde{V}_2$.

approximation to $\tilde{X}$ that can be reused across the models $i$ and $j$, $\tilde{E}_j \approx \tilde{E}_i * \tilde{V}_i^+ * \tilde{V}_j$. In this case, we overload notation from $\mathcal{T}_{i:j}$ and allow $\mathcal{T}$ to be a differentiable map between tensors of words, rather than tokens. Equation (2), can then be rephrased in terms of sets of whitespace-separated tokens, where 'merge($\mathcal{T}_{i:j}$(split($E_i$)))' is simply a mapping of an embedding from model $i$ to model $j$ in terms of our tensor-based vocabulary,

$$\underset{E_i}{\arg\min}\, \mathcal{L}_i(E_i) + \sum_{j \neq i} \mathcal{L}_j\bigg( \text{merge}(\mathcal{T}_{i:j}(\text{split}(E_i))) \bigg). \tag{4}$$

Every $\tilde{E}$ in split$(E) = \begin{pmatrix} \tilde{E}_1 & \tilde{E}_2 & \cdots & \tilde{E}_k \end{pmatrix}$ may have a potentially different length $l$, so if $\tilde{E}_1$ is the embedding for a model that tokenizes the word "Happy" with two tokens, {'Ha','ppy'} and $\tilde{E}_2$ has been constructed from a model that tokenizes it as a single token, {'Happy'}, we still need to ensure $\tilde{V}_i^+ * \tilde{V}_j$ are of appropriate sizes to compute the product. We can accomplish this by conditioning the mapping $\tilde{V}_i^+ * \tilde{V}_j$ on the length, $l$ of $\tilde{E} \in \mathbb{R}^{w \times d \times l}$, and keep track of specific $\tilde{V}_i^+ * \tilde{V}_j$ maps across 'sub'-vocabularies in which $V_j$ is comprised only of words that require $l$ tokens to represent. When computing the gradients, we simply check how many tokens each word requires and use the appropriate $\tilde{V}_i^+ * \tilde{V}_j$. See Algorithm 2 in Appendix B for a full description.

During a backward pass, we split the gradient from model $j$ into a set of tensors that have the same shape as calling 'split' on the original embeddings. We compute a final, approximate gradient by first converting model $i$'s embedding to text and then to model $j$'s embedding space, before computing the gradient of model $j$'s loss with respect to the correct embeddings. This gradient is then split apart and separated using the split operation and each piece is multiplied by the appropriate $\tilde{V}_i^+ * \tilde{V}_j$ based on its token length, before being merged back together into the appropriate gradient size for $E_i$ (see Figure 2 for a visualization and Algorithm 1 for pseudocode),

$$\nabla_{E_i} \mathcal{L}_j \bigg( \text{merge}(\mathcal{T}_{i:j}(\text{split}(E_i))) \bigg) \approx \text{merge}\bigg( (\tilde{V}_i^+ * \tilde{V}_j) * \text{split}(\nabla_{E_j}\mathcal{L}_j(E_j)) \bigg). \tag{5}$$

Just as in the case where we have the same tokenizer across models, this allows us to approximate the gradient across the tokenizers, enabling us to freely use gradient-based optimizers, while needing to store a set of parameters of size $d_i \times d_j \times \left( \sum_{i=1}^l i \right)$ tensor. In theory this $l$ could be very large, however, in practice we limit $l$ to a reasonable number, $l = 4$ as we expect the number of words that require more than 4 tokens to be fairly rare. For example, the Llama Tokenizer (Touvron et al., 2023) requires only 4 tokens to represent 97.6% of the text in the BookCorpus (Zhu et al., 2015) dataset.

---

**Algorithm 1:** Pseudocode for computing the FUSE Adapter backward pass.

---

**Input:** Gradient from model $j$: $\nabla_{x_j} f(x_j)$

**Input:** List of $(V_i^+ * V_j)$. List index corresponds to size of third tensor dimension

**Output:** Gradient w.r.t. model $i$'s embedding

```
1  L ← split(∇_{x_j}f(x_k))                        // Split gradient wrt each word
2  G ← empty list
3
   // For each word's gradient
4  for k ← length(L) do
5  |    m ← Sequence Length(L[k])                   // Tokens in this word
6  |    T ← (V_i^+ * V_j)[m]              // Index (V_i^+ * V_j) based on token count
7  |    G[k] ← L[k] * T                              // Compute Tensor Product
8  ∇_{x_i}f(𝒯_{i:j}(x_i)) = merge(G)                // Stack to matrix
9  return ∇_{x_i}f(𝒯_{i:j}(x_i))
```

---

# 4 Experiments

## 4.1 Datasets

We show that our approach effectively transfers knowledge across multiple models by focusing on two tasks: image captioning and image captioning with sentiment using the following datasets:

**MS-COCO (Karpathy Test Split) (Lin et al., 2014)** COCO provides 5000 images each with 5 human-annotated captions, allowing for the evaluation of image captioning quality.

**NoCaps-Val (Agrawal et al., 2019)** This dataset seeks to provide a more varied set of objects and concepts than included in MS-COCO. This dataset consists of 10600 test and 4500 validation images sourced from the Open Images (). Each image is accompanied by 10 human-annotated captions. The dataset is separated into an "in-domain", "near-domain", and "out-domain" splits that describe the degree to which the subset contains object classes common to MS-COCO images. Here we caption all images in the validation set.

**SentiCap (Mathews et al., 2016)** This dataset consists of 2360 images from the COCO Karpathy validation split, each with 6 new captions for each image, 3 positive sentiment captions and 3 negative sentiment captions. We use this dataset to investigate the ability to control the sentiment of a caption via a pretrained sentiment classifier.

## 4.2 Implementation Details

For the above datasets, we construct a simple captioner via a multi-objective optimization:

$$E^* = \arg\min_E \mathcal{L}_{CE}(f_\theta(E), E) + \alpha_1 \cdot \text{CLIP}_\theta(\mathcal{T}_{f:CLIP}(E), \mathcal{I}) + \alpha_2 \cdot \mathcal{L}_{CE}(g(\mathcal{T}_{f:g}(E)), s) \quad (6)$$

This equation minimizes the sum of the clip similarity between an image and the embedding, the cross entropy between this embedding and an arbitrary language model's output, and the correctness of the sentiment as determined by a BERT-based sentiment-classifier. Here $f$ is a pretrained language model (e.g., GPT2-Medium Radford et al. (2019)), $g$ is a sentiment classifier, $\mathcal{L}_{CE}$ is the cross-entropy loss, $\mathcal{T}_{f:CLIP}$ is the mapping from the language model's embeddings to CLIP's embeddings, $\mathcal{T}_{f:g}$ is the found mapping from the language model's embeddings to the sentiment classifier's embeddings, $s \in \{\text{positive}, \text{neutral}, \text{negative}\}$ is the desired sentiment, and $\alpha_i$ is a scalar weight. When captioning without sentiment, we set $\alpha_2 = 0$. In order to better compare with prior zero-shot methods, we use GPT2-Medium as our language model, and VIT-B/32 for CLIP and a Bert-based sentiment classifier[5].

---

[5]cardiffnlp/twitter-roberta-base-sentiment-latest

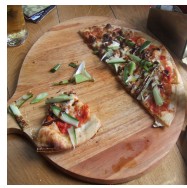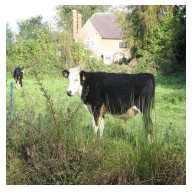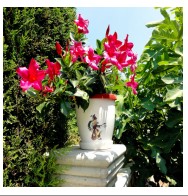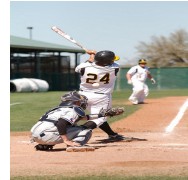

| | | | | |
|---|---|---|---|---|
| No Sentiment | A typical pizza from the Norfolk website. | A cow grazing on a patch of bush close to where she lived. | A flower pot in the garden of the terrace house. | A player hitting a home run. Photo: Sierra Vista College. |
| Positive | A pizza made with organic ingredients. Photo: Fairfax | A cow grazing on a hedge in front of the village. | One of the flowers stands on a pot in a garden outside a house in | The ball hitting the back of the built-in sliding bat. Note the |
| Negative | A pizza being served to a group of students demonstrated how widespread this behaviour is. | A cow in front of a ditch in the southeast countryside. | A white bucket with a red flower on it has been | A man hitting and stomping on a college senior |

Figure 3: Example Captions that using a FUSE Adapter to minimize the sum of GPT2-Medium, CLIP-VIT-B/32, and a Bert-based Sentiment Classifier via AutoDAN (Zhu et al., 2023). This combination of models controls through synonyms that indicate tone or through creating additional context for each image to denote tone. Note that AutoDAN does not have a clear stopping condition, a caption may stop in the middle of a sentence.

When fitting FUSE, we limit it to computing gradients of words that require 4 or fewer tokens. We fit the adapter using 16384 random words from the Wiki-Text dataset for each case where words require less than 4 tokens as described in Section 3.3 and Algorithm 2. If a word uses more than 4 tokens to represent, we treat the Jacobian used by FUSE as a random matrix, expecting further optimization steps to insert a token with white-spacing, reverting to the setting that the adapter is fit to. Fitting the adapter for the models considered in our experiments requires only 4 minutes and 22 seconds on a standard workstation with 32GB of memory. As shown in Figure 2, during optimization, the forward pass consists of a mapping from embeddings to text and back again, limited only by the time required to perform this mapping. During the backward pass we only require a single t-product, which consists of the sum of $m^2$ matrix multiplications, where $m$ is the number of tokens that make up each word.

We then use the discrete optimizer AutoDAN Zhu et al. (2023) to optimize the objective. In contrast to methods like, (Zou et al., 2023) and (Wen et al., 2024), AutoDAN optimizes a prompt one token at-a-time by first computing the log probabilities of the next token using our given language model and some prefix, and adds these logits to the negative gradient of the objective. This sum returns a set of scores that describe an estimate of the improvement in the loss for each token. We choose the top 512 candidates and compute the true error to determine the best token update. Unlike AutoDAN, which performs this search greedily, we also use a beam search with a beam width of 5 when searching through the space of token updates. All captions use the prefix "An image of" at initialization.

We assess the FUSE Adapter's performance for image captioning using standard supervised metrics: BLEU-N (Papineni et al., 2002), METEOR (Banerjee & Lavie, 2005), CIDEr (Vedantam et al., 2015), SPICE (Anderson et al., 2016) that measure caption quality against human-written references, evaluating captions for n-gram overlap (BLEU-N), semantic similarity (METEOR), content alignment (CIDEr), and grammatical coherence (SPICE).

## 5 Results

### 5.1 Image Captioning

In Table 1, we show our results on MS-COCO and NoCaps-Val. As with other zero-shot captioning methods, without domain bias for human captions, we do not expect that we will be able to achieve the same level of performance as models that have been finetuned for captioning. However, among zero-shot methods, our approach significantly improves

| Metrics | MS-COCO | | | | NoCaps-Val (Overall) | |
|---|---|---|---|---|---|---|
| | B-4 | M | C | S | C | S |
| **Supervised Methods** | | | | | | |
| BLIP-2 (Li et al., 2023) | 43.7 | - | 145.8 | - | **119.7** | **15.40** |
| mPLUG (Li et al., 2022) | **46.5** | 32.0 | **155.1** | 26.0 | 114.8 | 14.8 |
| OFA (Wang et al., 2022) | 44.9 | **32.5** | 154.9 | **26.6** | - | - |
| CLIP-VL (Tewel et al., 2021) | 40.2 | 29.7 | 130.3 | 23.8 | - | - |
| VinVL (Zhang et al., 2021) | 40.9 | 30.9 | 140.4 | 25.1 | 90.4 | 13.07 |
| LEMON-B (Hu et al., 2022) | 40.3 | 30.2 | 133.3 | 23.3 | 79.0 | 12.3 |
| ClipCap (Mokady et al., 2021) | 32.2 | 27.1 | 108.35 | 20.12 | 65.7 | 11.1 |
| **Zero Shot Methods** | | | | | | |
| ZeroCap (Tewel et al., 2022) | **2.60** | 11.50 | 14.60 | 5.50 | - | - |
| ConZIC (Zeng et al., 2023) | 1.29 | 11.23 | 13.26 | 5.01 | - | - |
| Ours ( GPT2-M + VIT-B/32 ) | 1.59 | **14.72** | **15.93** | **9.15** | 20.65 | 6.64 |

Table 1: Comparison of SOA image captioning methods.

among most of our metrics. Moreover, we see a significantly larger increase in the SPICE score over our zero-shot comparison methods; our caption generation process returns more grammatically consistent text as the comparisons. This is likely due to using AutoDAN as our discrete optimizer, which places weight on not just the objective but the direct probabilities of each new token before computing the cross-entropy over GPT2-M's logits. As our discrete optimizer determines candidates based on the gradient of Equation (6), the observed performance necessitates that the gradient of the CLIP similarity between the image and the CLIP's text embeddings, with respect to GPT2-M's text embedding is meaningful.

## 5.2 Captioning with Sentiment

Table 2 shows our method's performance on image captioning with sentiment. As in the standard captioning task above, we see that combining CLIP-VIT-B/32, GPT2-M, and a Bert-based sentiment classifier, successfully finds a caption that aligns well with the semantic content of the reference. But, we are less accurate in the found sentiment than the comparison methods. While most methods insert descriptive adjectives that denote sentiment, at every step we are trying to minimize both the image similarity and the sentiment. As a result, our approach finds synonyms that connote the sentiment. For example, in Figure 3, a negative caption replaces the "flower pot" with "bucket". In the context of a replacement word for 'flower pot' bucket carries a more negative sentiment, however, at face value, "a bucket with a red flower" is a neutral statement. Again, our results are not focused on improving over other methods in terms of performance on such datasets, but showing that the FUSE Adapter provides meaningful gradients in its backward pass. The changes to the standard captions elicited by the BERT-based sentiment classifier also necessitate that each gradient step is carrying information from both the image and sentiment.

## 6 Conclusion and Future Work

In this work, we propose a novel approach for approximating gradients across models and tokenizers during prompt optimization. We introduce an adapter that precisely maps across token and embedding spaces in the forward pass. By leveraging a precomputed linear transformation, we efficiently approximate the behavior of a true differentiable mapping between embedding spaces during the backward pass. This adapter not only improves accessibility for knowledge transfer tasks for prompt optimization, but also unlocks potential new tasks by allowing for easy compositions of distinct models.

We demonstrate the potential of our approach on zero-shot image classification tasks, where combining a language model, a vision-language model, and a Bert-based sentiment classifier in a multi-objective optimization, we achieve superior results to prior zero-shot image captioning methods. This suggests that despite being an approximation our gradient carries meaningful information.

| Metrics | Positive | | | Negative | | |
|---|---|---|---|---|---|---|
| | B-3(↑) | M(↑) | Acc(↑) | B-3(↑) | M(↑) | Acc(↑) |
| **Supervised** | | | | | | |
| StyleNet (Gan et al., 2017) | 12.1 | 12.1 | 45.2 | 10.6 | 10.9 | 56.6 |
| MSCap (Guo et al., 2019) | 16.2 | 16.8 | 92.5 | 15.4 | 16.2 | 93.4 |
| MemCap (Zhao et al., 2020) | 17.0 | 16.6 | 96.1 | 18.1 | 15.7 | **98.9** |
| ADS-CAP (Cheng et al., 2022) | **18.9** | **18.5** | **99.7** | **21.0** | **18.0** | 98.2 |
| **Zero Shot** | | | | | | |
| ConZIC (Zeng et al., 2023) | 1.89 | 5.39 | **97.2** | 1.78 | 5.54 | **99.1** |
| Ours ( GPT2-M + VIT-B/32 + Roberta) | **1.91** | **10.40** | 83.8 | **2.29** | **7.42** | 85.6 |

Table 2: Comparison of SOA sentiment-based image captioning methods.

While this work introduces a simple adapter, researchers and organizations may prefer learning an actual mapping through supervised learning of a transformer to translate from one embedding space to another. Yet, the compute necessary for such a task may not be universally available. We believe that FUSE may serve a valuable purpose in low-resource/low-compute settings, in which researchers may want to do inference across models, yet be unable to train a true adapter. Additionally, this approach may be useful in fast-paced environments, where FUSE can be used as a low-cost preliminary test for more involved methods requiring a well-trained adapter.

Our work presents an initial step to making prompt optimization more accessible and scalable. Future research may explore more memory and storage-efficient approaches while improving upon the accuracy of our proposed method. Since this work approximates a differentiable map from one discrete space to another, it is important to note that the traditional concept of a gradient does not apply, as such traditional ways of validating gradient approximations were unavailable. Future work may introduce comprehensive validation methods for mappings and gradients from one discrete embedding space to another. Our work also opens the door for further investigations of techniques that mitigate the storage costs associated with longer sequences and integrating more advanced mapping approximations. While there remain areas to build on, our approach holds promise for improving methods of prompt optimization, particularly in resource-constrained settings, and lays the groundwork for future innovations in cross-model interactions.

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

# A    Tensor Products

Recall that the order (aka. modes or ways) of a tensor is the number of dimensions that make it up. Kolda & Bader (2009) have used one dimensional fibers or two dimensional slices to define tensors, where a third-order rank one tensor is defined as,

$$\tilde{A} = a \circ b \circ c,$$

where $\circ$ denotes the outer product operation between vectors $a$ and $b$, defined as

$$a \circ b = \begin{bmatrix} a_0 b_0 & \cdots & a_0 b_n \\ a_1 b_0 & \cdots & a_1 b_n \\ \vdots & \ddots & \vdots \\ a_n b_0 & \cdots & a_n b_n \end{bmatrix} \qquad A \circ b = \begin{bmatrix} A_0 b_0 & A_1 b_1 & \cdots & A_n b_n \end{bmatrix}$$

Multiplication between tensors has been introduced in Kilmer & Martin (2011), in terms of the ciruclant matrix, where,

$$a = \begin{bmatrix} a_0 & a_1 & a_2 & a_3 \end{bmatrix}^T$$

then

$$\text{circ}(a) = \begin{bmatrix} a_0 & a_3 & a_2 & a_1 \\ a_1 & a_0 & a_3 & a_2 \\ a_2 & a_1 & a_0 & a_3 \\ a_3 & a_2 & a_1 & a_0 \end{bmatrix}.$$

In order to multiply tensors, we first, we define an unfolding operation that reshapes an $\mathbb{R}^{d_1 \times d_2 \times \cdots \times d_n}$ tensor into a partitioned tensor in $\mathbb{R}^{d_1 d_n \times \cdots \times d_{n-1}}$ tensor and we conversely define a fold operation to reshape the tensor back into its original shape,

$$\text{unfold}(\tilde{X}) = \begin{bmatrix} \tilde{X}_1 & \tilde{X}_2 & \cdots & \tilde{X}_n \end{bmatrix}^T \qquad \text{fold}(\text{unfold}(\tilde{X})) = \tilde{X}. \tag{7}$$

Using this notation, Kilmer & Martin (2011) defines the t-product between tensors recursively as,

$$\tilde{A} * \tilde{B} = \text{fold}(\text{circ}(\text{unfold}(\tilde{A}))) * \text{unfold}(\tilde{B}))$$

$$= \text{fold}(\begin{bmatrix} \tilde{A}_0 & \tilde{A}_1 \\ \tilde{A}_1 & \tilde{A}_0 \end{bmatrix} * \begin{bmatrix} \tilde{B}_0 \\ \tilde{B}_1 \end{bmatrix})$$

$$= \text{fold}(\begin{bmatrix} \tilde{A}_0 * \tilde{B}_0 + \tilde{A}_1 * \tilde{B}_1 \\ \tilde{A}_1 * \tilde{B}_0 + \tilde{A}_0 * \tilde{B}_1 \end{bmatrix}),$$

where circ is the circulant matrix. It is well known that the circulant matrix has a strong connection to circular convolutions as shown in Bamieh (2018). We can thus think of the t-product as a convolution with circular padding,

$$\tilde{A} * \tilde{B} = \begin{bmatrix} \tilde{A}_0 & \tilde{A}_1 \end{bmatrix} \otimes \begin{bmatrix} \tilde{B}_0 & \tilde{B}_1 & \tilde{B}_0 \end{bmatrix},$$

where $\otimes$ denotes a convolution of $\tilde{A}$ across $\tilde{B}$, using the t-product instead of the matrix multiplication. In this way, we can express a generalization of the t-product. Kilmer & Martin (2011) defined the t-product in terms of, $\tilde{A} \in \mathbb{R}^{m \times k \times p_1 \times \cdots p_n}$ and $\tilde{B} \in \mathbb{R}^{k \times n \times p_1 \times \cdots p_n}$, where $\tilde{A}$ and $\tilde{B}$ must have their first two dimensions of the appropriate shape for matrix multiplication and each of the remaining dimensions must be the same size for both tensors.

As a circular convolution, we can allow arbitrary tensor products as long as the tensors are of the same order by applying circular padding. For example, if $\tilde{A} \in \mathbb{R}^{m \times k \times 2}$ and $\tilde{B} \in \mathbb{R}^{k \times n \times 4}$, we can express the product as,

$$\tilde{A} * \tilde{B} = \begin{bmatrix} \tilde{A}_0 & \tilde{A}_1 \end{bmatrix} \otimes \begin{bmatrix} \tilde{B}_0 & \tilde{B}_1 & \tilde{B}_2 & \tilde{B}_3 & \tilde{B}_0 \end{bmatrix} \in \mathbb{R}^{m \times n \times 4}$$

Note that this product is equivalent to that described in Kilmer & Martin (2011) when $\tilde{A}$ and $\tilde{B}$ have the same sized dimensions after dimension 2. Moreover, it is easy to verify that this generalization still follows the same rules of distributivity and associativity as the standard t-product.

## B  Precomputing the Gradient $V_i^+ V_j$

---

**Algorithm 2:** Precomputing the Gradient $V_i^+ V_j$ for words that are tokenized to $l$ tokens

**Input:** Text Corpus $C$, Language models $\mathcal{M}_i$ and $\mathcal{M}_j$

**Output:** Gradient $V_i^+ V_j$

1   $l \leftarrow$ only consider words that require $l$ tokens in $\mathcal{M}_j$;

2   $\mathcal{T}_i, \mathcal{T}_j \leftarrow$ Tokenizer of model $\mathcal{M}_i, \mathcal{M}_j$;

3   $\mathcal{E}_i, \mathcal{E}_j \leftarrow$ Mapping from token to embedding of $\mathcal{M}_i, \mathcal{M}_j$;

4   $d_i, d_j \leftarrow$ Dimensionality of $\mathcal{M}_i, \mathcal{M}_j$ embeddings;

5   $W \leftarrow \varnothing$;                 // Initialize an empty list

6   $k \leftarrow 0$;            // keep track of max size to tokenize with $\mathcal{T}_i$

7   **foreach** *word in C* **do**

8      **if** *word $\notin W$* **then**

9          $t_j \leftarrow T_j(word)$;           // Tokenize a single word

10         $k \leftarrow \max(k, |T_i(word)|)$;          // update $k$

11         **if** $|t_j| = l$ **then**

12            $W \leftarrow W \cup \{word\}$;     // Add to list if exactly $l$ tokens in $j$

13   $V_i \leftarrow$ initialized zero tensor of $|W|$ rows, $d_i$ columns, and depth $l$;

14   $V_j \leftarrow$ initialized zero tensor of $|W|$ rows, $d_j$ columns, and depth $k$;

15   **for** $m \leftarrow 1$ **to** $|W|$ **do**

16      $t_j \leftarrow T_j(W[m])$;         // Tokenize word $W[m]$ with tokenizer $j$

17      $t_i \leftarrow T_i(W[m])$;

18      **for** $n \leftarrow 1$ **to** $|t_j|$ **do**

19        $(V_j)_{w,:,m} \leftarrow (V_j)[m,:,n] + \mathcal{E}_j((t_j)[n])$;    // Add the embedding of $t_j$ to $V_j$

20      **for** $n \leftarrow 1$ **to** $|t_i|$ **do**

21        $(V_i)_{w,:,m} \leftarrow (V_i)[m,:,n] + \mathcal{E}_i((t_i)[n])$;    // Add the embedding of $t_i$ to $V_i$

22   $V_i^+ \leftarrow$ Pseudoinverse( $V_i$ );        // According to Jin et al. (2017)

23   $\nabla_{E_i} T_{i:j}(E_i) \leftarrow V_i^+ * V_j$;          // Compute the t-product

24   **return** $\nabla_{E_i} T_{i:j}(E_i)$

---

