# OpenReview forum: "FUSE-ing Language Models: Zero-Shot Adapter Discovery for Prompt Optimization Across Tokenizers"
_colmweb.org/COLM/2024/Conference — COLM_

### Official Review · Reviewer_cRDb · 2024-05-05

**Rating:** 7
**Confidence:** 4
**Ethics Flag:** 1

**Summary:**

Recently, with the increasing research on LLMs, there is a diversity of LLMs, each with its own unique tokens and embedding spaces. This diversity has led to the encapsulation of knowledge and understanding within these models, posing challenges for knowledge transfer and collaboration between models. To address this challenge, one current approach is to utilize prompt design to facilitate knowledge transfer across various models through text. However, the effectiveness of this method is still constrained by the uniqueness of the token and embedding spaces between models. In response to this, this paper proposes FUSE (Flexible Unification of Semantic Embeddings), which maps the disparate embedding spaces between two LLMs without the need for fine-tuning. The paper also conducts relevant experiments, showing a considerable improvement, particularly with a nearly 50% enhancement in the SPICE evaluation metric. This paper offers a new, cost-effective method for cross-model collaboration and knowledge transfer, making a significant contribution to LLMs research.

**Questions To Authors:**

1.On the second page, in the last paragraph of the Introduction section, it is suggested to list the contributions of this paper from most to least important.

2.On the second page, in the Background and Related Work section, it is recommended to add subheadings to make the discussion clearer and more organized.

3.On the fifth page, it is suggested to adjust the layout of Figure 2 for better readability, preferably in a vertical arrangement.

4.On the eighth page, in the Experiments section, in Table 1, the proposed method from this paper is not compared with other methods in the Supervised Methods category. It's suggested to add experiments for comparison.

**Reasons To Accept:**

The acceptance of this article can mainly be attributed to the following reasons:
1. Innovative approach: The paper introduces a novel method called FUSE (Flexible Unification of Semantic Embeddings), which maps the disparate embedding spaces between two LLMs without the need for fine-tuning. The innovation lies in addressing the challenge of knowledge transfer between LLMs while also providing a cost-effective and efficient solution.

2. Contribution to LLMs research: By proposing a new method and validating it through experiments, this paper makes a contribution to the research on LLMs. This contribution lies in offering an innovative solution to the challenge of collaboration and knowledge transfer between LLMs, thereby advancing the field.

**Reasons To Reject:**

1.Image layout needs improvement: Images are crucial supplementary materials in research articles. However, if the layout is unclear or not standardized, it may hinder readers' understanding and absorption of the article's content. Reviewers might consider poor image layout as detrimental to the readability and attractiveness of the article.

2.Lack of comparison with other methods: In the field of research, newly proposed methods typically require comparison with existing methods to validate their effectiveness and superiority. If the article lacks comparative experiments or analysis with other methods, reviewers may perceive that the authors have not adequately demonstrated the advantages or innovation of their method.

---

> ### Author Rebuttal · Authors · 2024-05-29
>
> Thank you for your thoughtful response. Regarding the questions raised:
>
> 1) At the end of our introduction, we currently include a list of contributions that we feel are in the order of most to least important
> 2) We have updated the background/related works section to have additional subheading to assist with readability
> 3) Regarding the readability issues in figure 2, can you provide more clarification on the issues with the figure? Due to space constraints a vertical layout may be difficult, especially as we aimed to have the text within the figure be about the size of the main-text to ensure readability. If accepted, COLM allows for an additional page of space in the camera ready, we may then be able to explore a vertical layout for this figure.
> 4) Regarding the comparison to other methods, our method is not supervised, so we cannot add additional supervised experiments. The layout of Table 1 aligns with the layout of other articles on zero-shot captioning by separating Supervised and Zero-shot methods to show the relative improvements across methods.
>
> Regarding the comparison to existing methods, to the best of our knowledge there are no other methods that look at computing gradients across tokenizers. If you are familiar with any, please let us know and we will update the experiments to reflect this in future revisions. As we are not aware of methods that we can directly compare to, we focus on showing how our framing and method can be used to improve on existing methods, if those methods can be expressed as minimizing the sum of a readability and image-text alignment objectives or other aggregates of LLMs and MLLMs.

---

### Official Review · Reviewer_uQbT · 2024-05-10

**Rating:** 7
**Confidence:** 3
**Ethics Flag:** 1

**Summary:**

This paper proposes an approach for converting adapter layers from one tokenization to another. Two models might have different embedding matrices and even tokenizations (ie different ways of splitting words into subwords). This paper proposes a tensor product approach to handle the alignment problem between tokenizations of different lengths. The experiments suggest that on an image captioning task, Bert-BASE can supervise a GPT2-M based captioner (with different tokenizations).

---
update post rebuttal: thanks authors! I don't think anything mentioned changes my judgment here, so I'll keep my score.

**Reasons To Accept:**

* Seems like an interesting approach for handling the alignment problem between tokenizers. I'm not sure if other people in the community have thought of these issues.
* The results show that this approach is able to translate gradients from one model to another model (with a different tokenizer). It's not clear to this reviewer that this approach is necessarily a good one, but it at least seems to do something :)


Overall, this reviewer is in favor of accepting because it can maybe shed light on some of these the inner workings around embeddings, soft tokens, etc!

**Reasons To Reject:**

The main concern of this reviewer is that this approach is pretty complicated, and the complicatedness might diminish its utility. Likewise, it seems to hinge on the fact that tokenizers that people commonly use (e.g. GPT2 versus BERT) mostly have the same words in common, and split on whitespace. The application space also feels a bit narrow to this reviewer (feels a bit toy to assume a language only model and that we want to somehow adapt in captioning ability in this way, instead of say, finetuning on caption data)

---

> ### Author Rebuttal · Authors · 2024-05-29
>
> Thank you for your thoughtful comments. You are correct that this approach hinges on the need for tokenizers to split on whitespace. While  some tokenizers could break apart words differently, in practice most current tokenizers focus on whitespace splitting. Moreover, our approach does not require tokenizers to have the same words in common, only that tokenizers can build the same word out of the tokens in their vocabulary. The method is agnostic to the model/tokenizer used. We chose CLIP and GPT2 here in order to both align with existing zero-shot methods for our analysis, and to illustrate a case in which we have very different tokenizers.
>
> It is also for this reason that we include sentiment-based captioning. By combining 3 very different types of models, we believe that we show that we find a meaningful jacobian across combinations of LLMs, MLLMs, and simpler classifiers – any model that accepts natural language as input.
>
> And while one could design a better captioner by finetuning existing models, as we say in Section 5.2: “our results are not focused on improving over other [captioners] in terms of performance…, but showing that the FUSE Adapter provides meaningful gradients in its backward pass.” While some organizations may not need this sort of approach, we believe that this could open doors for low-resource/low-compute users who may not have the resources to finetune or learn mappings across models.  We have added a bit to our conclusion to highlight this.
>
> Lastly, regarding your concern on the complexity of the approach and the nicheness of its applications, as you say, our goal here was to fill something into this space where we may want to optimize across tokenizers, even if this approach has the potential for further development. We hope that this work could assist in our understanding of hard/soft embeddings, especially as they pertain to the ways that text is handled across models, tokenizers and architectures

---

### Official Review · Reviewer_jJEH · 2024-05-11

**Rating:** 6
**Confidence:** 3
**Ethics Flag:** 1

**Summary:**

This paper proposes a way to do gradient based update to one model's output wrt another model (with a different tokenizer)'s embedding space. The proposal is a simple adapter layer used to map between the embeddings of models without fine-tuning. Using some linear algerbra, they show how gradients can be approximated using their formulation.

**Questions To Authors:**

1. Is it possible to study the accuracy of gradient approximation using empirical metrics?

**Reasons To Accept:**

1. The high-level motivation is novel i.e. to be able to compute forward and backward passes across models with different tokenization.
2. The positive results on MS-COCO with metrics such as B-3, METEOR, CIDER are noteworthy.

**Reasons To Reject:**

1. As the authors noted, the gradients are approximate, so the model may underperform other baselines.
2. The application of the proposed approach is rather niche. Why do you need to choose a language model with a different tokenization when you can use a similarly performing model with the same tokenization?
3. Writing issues: typos (Fig 1 caption, page 6 last paragraph), missing citations (page 7)

---

> ### Author Rebuttal · Authors · 2024-05-29
>
> Thank you very much for your thoughtful comments, and for pointing out the typo/missing citation. We have updated these in the draft.
>
> Regarding your question about the accuracy of the gradient approximation using empirical metrics, it is unclear how to measure the accuracy of the gradient in this way. As we map from one discrete space to another discrete space, the concept of a traditional gradient does not directly apply, and as such, standard ways of measuring the accuracy of the gradient using empirical measures are unavailable. We have looked at validating that the backward pass returns something that acts in the way that we expect a gradient to, e.g.,  if we map from model A to model B and back to model A, the gradient of the embeddings should be the identity matrix. However, this would only be true in the case where we dealt with the true inverse and not the pseudoinverse. From our preliminary tests this metric was very inconclusive, so we focus on the experimental results in the text.
>
> Regarding your question about using a language model with a different tokenization, rather than a model with a similar tokenization, we disagree that this is a niche case. There are several applications where a similarly performing model may not be available, e.g., to the best of our knowledge there are no text-generation models that use clip embeddings/tokenizers. And while some organizations may be able to train their own models to fit this niche, we believe that this approach opens the door for low-resource/low-compute users to do more complex tasks. We have updated our conclusion to highlight this case.

---

> > ### Comment · Reviewer_jJEH · 2024-06-05
> > **How to study the accuracy of gradient approximation**
> >
> > Regarding the accuracy of the gradient approximation, have you consulted any experts in linear algebra to check any way to achieve this?
> > " identity matrix. However, this would only be true in the case where we dealt with the true inverse and not the pseudoinverse. " - I would advise the authors to get to the bottom of this since this papers delves a lot on theoretical foundations of such a mapping. It is not sufficient to say that "From our preliminary tests this metric was very inconclusive"- this might demonstrate a shortcoming of the current approach but please add those results atleast.

---

> > > ### Author Response · Authors · 2024-06-05
> > >
> > > Thank you for following up on this point. Sorry if we were not clear, but the point we are trying to make isn’t that the metric we mentioned is somehow highlighting any ambiguity with the approach (“inconclusive” was a poor word choice), but rather that the metric is simply not a very relevant way to evaluate the quality of the approximation. To clarify the issue here more formally, taking the first case, where both models share a tokenizer:
> > >
> > > Let $f( y )$ be the map from model A’s embeddings to model B’s embeddings, and $g( x )$ be the map from model B’s embeddings to model A’s embeddings. Since these are maps across discrete spaces, the traditional concept of a gradient does not apply. We therefore briefly considered a metric that looks at the behavior of the FUSE gradient. This metric was based on the similarity between the identity matrix and the jacobian of the map from model A’s embeddings to model B’s embeddings and then back to model A’s embeddings:
> > >
> > > $J_{f \circ g}( x ) = I$.
> > >
> > > This can easily be evaluated as: $J_{f}( y ) J_{g}( x )$. Where $J_{f}$ and $J_{g}$ are the Jacobians of f and g respectively. In section 3.1, we propose an approximation to $J_{f}$ and $J_{g}$ as $V_{B}^{+} V_{A}$ and $V_{A}^{+} V_{B}$ respectively, where $V_{A}$ and $V_{B}$ are the discrete input embedding spaces of model A and model B. We can then show the following about the Jacobian:
> > >
> > > $J_{f \circ g}( x ) = J_{f}( y ) J_{g}( x ) = V_{B}^{+} V_{A} V_{A}^{+} V_{B}$.
> > >
> > > This equation is the identity matrix if we can take the true inverses $V_{B}^{-1}$ and $V_{A}^{-1}$. However, one of the properties of the pseudoinverse is that if $A A^+ = I$ then $A^+ A \neq I$ or the other way around depending on the size of the row/column spaces. Therefore:
> > >
> > > $V_{B}^{+} V_{A} V_{A}^{+} V_{B} = V_{B}^{+} V_{B}$.
> > >
> > > This is not equal to the identity matrix. As $V_{B}$ approaches a full-rank matrix, the pseudoinverse becomes more like a true inverse, and $V_{B}^{+} V_{B}$ becomes closer to the identity. We know ahead of time that the embedding matrices are not full rank, so $J_{f \circ g}( x )$ will never be the identity.
> > >
> > > Measuring the quality of the approximation in this way becomes a function of the ranks of $V_{A}$ and $V_{B}$. This metric would effectively be reporting the ranks of the embedding matrices, which we do not believe provides useful information when determining the quality of the gradient approximation.

---

### Official Review · Reviewer_1ZFn · 2024-05-13

**Rating:** 4
**Confidence:** 4
**Ethics Flag:** 1

**Summary:**

The paper introduces FUSE (Flexible Unification of Semantic Embeddings), a new method designed to facilitate prompt optimization and knowledge transfer across large language models with diverse tokenizers and embedding spaces. Addressing the challenge of semantic alignment in multi-model environments, FUSE employs a third-order tensor-based representation to align textual embeddings that have been disjointedly processed by different tokenizers. This approach enables the computation of gradients through embedding spaces of multiple models, thereby approximating an adapter layer that can effectively bridge gaps between these models. The effectiveness of FUSE is demonstrated through multi-objective optimization tasks, including vision-language and causal language models for applications such as image captioning and sentiment-based image captioning.

**Questions To Authors:**

Please check the reason to reject for details.

**Reasons To Accept:**

1. The proposed approach is intuitive.
2. Some of the results look good.
3. The idea is overall well presented.

**Reasons To Reject:**

1. The reviewer has a significant concern on the setup of the current work. The paper targets at zero-shot adapter across tokenizers but the current setting is limited to bridge a vision-language model and language models. But what is more exciting and important part under this hood should be adapt two LLMs or MLLMs. The current setup seems less practical and important considering the existing strong MLLMs that can do image captioning with much better quality.

2. The authors mentioned in abstract that the proposed method is inexpensive but it is unclear what is the training/inference cost of the method.

3. The current experiment setting is also in a rather smaller scale and low-performance regime. What if scaling up the models used?

---

> ### Author Rebuttal · Authors · 2024-05-29
>
> Thank you very much for your thoughtful comments. Regarding your concern about limited scope, while the experiments here are focused on combining VLMs and LMs, our approach is model agnostic. As we say in Section 5.2: “our results are not focused on improving over other methods in terms of performance… but showing that the FUSE Adapter provides meaningful gradients in its backward pass.” Combining a VLM and an LM is one clear way to do so.
>
> On a fundamental level, our technique is about bridging the tokenizer aspects of different models. Tokenizers are fundamental to modern LLMs and MLLMs, and despite some efforts in the area, there does not seem to be a consensus emerging around a “single” set of tokens (recent models all use their own custom tokenizers built from their own training data). FUSE represents a mechanism by which pre-trained models can be combined in an extremely cheap manner, whereas retraining LLMs and MLLMs from scratch would be infeasible. We focused on the smaller scale/low-performance regime in order to compare to existing zero-shot methods which are also in this regime.
>
> Regarding the efficiency of our approach, we have updated our implementation details to give a better sense here. For the case in which two models share the same tokenizer, we only need to precompute the pseudo-inverse of one of the vocabularies, which has complexity O( d^2 |v| ), where the embedding is d-dimensional with |v| unique tokens. In the more complex case, in which the models have different tokenizers, the pseudo-inverse of a third order tensor requires two applications of the fast fourier transform and m pseudoinverses, where m is the depth of the rank 3 tensor. This gives a complexity of O( 2 ( N ) log( N ) + N * d ), where N is the number of elements in the rank 3 tensor of words/tokens. (See algorithm 5.1 of [1] ). In practice, on a standard workstation, our experiments required 4 minutes and 22 seconds to fit the adapter over 16384 words.
>
> The inference cost of the method is limited only by the amount of time required to convert to and from text. The backward pass is only the time required to compute the t-product, which requires m^2 d-dimensional matrix multiplications m are the number of tokens required to represent a word (In our experiments m <= 4).
>
> [1] Jin, Hongwei, et al. "The generalized inverses of tensors and an application to linear models." Computers & Mathematics with Applications 74.3 (2017): 385-397.

---

### Decision · Program_Chairs · 2024-07-10

**Decision:**

Accept

**Comment:**

This paper proposes an approach for mapping the semantic embedding of two models trained with different tokenizers in a zero-shot way. More specifically, the authors first map the semantic embedding of models (with different tokenizers), and then derive an approximate gradient of one model's output with respect to the other model's input. As a result, they discover an adapter in a zero-shot way. I agree with the reviewers on the innovativeness of the approach but also highlight their feedback on evaluation in different settings: the work would be stronger and more impactful if it is evaluated in (1) other settings like mapping two LLMs (2) more benchmarks in the VLM study (3) low resource settings--especially, if the main use case of this approach is for cases where fine-tuning data is not available, e.g., low-resource languages. Moreover, while it is not possible to compare this method directly with the existing supervised adaptation approaches, the authors can consider such approaches as an upper-bound--discuss how far from the upper-bound is your current results. Also, you could try a supervised method but increasing the amount of fine-tuning data, and see what amount of data is needed to achieve your results. I strongly encourage the authors to address these comments for the next revision of the paper.